# SISPO: Space Imaging Simulator for Proximity Operations

**Mihkel Pajusalu**[1][◉], **Iaroslav Iakubivskyi**[1][◉]*, **Gabriel Jörg Schwarzkopf**[2][◉], **Olli Knuuttila**[2][◉], **Timo Väisänen**[2][◉], **Maximilian Bührer**[2,3][◉], **Mario F. Palos**[2][◉], **Hans Teras**[1][‡], **Guillaume Le Bonhomme**[1][‡], **Jaan Praks**[2][‡], **Andris Slavinskis**[1][‡]

**1** Space Technology Department, Tartu Observatory, University of Tartu, Tõravere, Estonia, **2** Department of Electronics and Nanoengineering, School of Electrical Engineering, Aalto University, Espoo, Finland, **3** School of Aerospace, Transport and Manufacturing, Cranfield University, Cranfield, Bedfordshire, United Kingdom

◉ These authors contributed equally to this work.
‡ HT, GLB, JP and AS also contributed equally to this work.
* iaroslav.iakubivskyi@ut.ee

**Data Availability Statement:** All SISPO algorithm files are available from the GitHub (https://github.com/SISPO-developers).

**Funding:** This work was funded by the ESA Contract No. 4000131003/20/NL/IB/ig with the

## Abstract

This paper describes the architecture and demonstrates the capabilities of a newly developed, physically-based imaging simulator environment called SISPO, developed for small solar system body fly-by and terrestrial planet surface mission simulations. The image simulator utilises the open-source 3-D visualisation system Blender and its Cycles rendering engine, which supports physically based rendering capabilities and procedural micropolygon displacement texture generation. The simulator concentrates on realistic surface rendering and has supplementary models to produce realistic dust- and gas-environment optical models for comets and active asteroids. The framework also includes tools to simulate the most common image aberrations, such as tangential and sagittal astigmatism, internal and external comatic aberration, and simple geometric distortions. The model framework's primary objective is to support small-body space mission design by allowing better simulations for characterisation of imaging instrument performance, assisting mission planning, and developing computer-vision algorithms. SISPO allows the simulation of trajectories, light parameters and camera's intrinsic parameters.

## Introduction

A versatile image-simulation environment is required in order to design advanced deep-space missions, to simulate large sets of mission scenarios in parallel, and to develop and validate algorithms for semi-autonomous operations, visual navigation, localisation and image processing. This is especially true in the case of Small Solar System Body (SSSB) mission scenarios, where the mission has to be designed with either very limited information about the target (i.e., precise size, shape, exact composition and activity) or the targets can remain a near-complete mystery before their close encounter (i.e., as in the case of interstellar objects [1, 2]). Some publicly known cosmic-synthetic-image generators for space missions are available, and they are briefly described in the next paragraph.

University of Tartu ("Comet Interceptor (EE-1): OPIC Engineering Model Development", PI is MP; the salary was paid to: MP, II, GLB, HT, and hardware acquisition), the Archimedes Foundation (https://archimedes.ee, UT ASTRA project 2014–2020.4.01.16-0029 KOMEET "Benefits for Estonian Society from Space Research and Application" in the form of travel supports), the Eesti Teadusagentuur (EE) (MOBTP151 and PUTJD601, awarded to MP), and the base funding of Tartu Observatory (registration number 74001073). The funders had no role in study design, data collection and analysis, decision to publish, or preparation of the manuscript.

**Competing interests:** The authors have declared that no competing interests exist.

**Abbreviations: 3-D**, three-dimensional; **BRDF**, Bidirectional Reflectance Distribution Function; **ESA**, European Space Agency; **JAXA**, Japan Aerospace Exploration Agency; **MAT**, Multi-Asteroid Touring; **OASIS**, Optical Aberrations for Still Images Simulator; **OpenGL**, Open Graphics Library; **OpenMVG**, Open Multiple View Geometry; **OpenMVS**, Open Multiple View Stereo Reconstruction; **OPIC**, Optical Periscopic Imager for Comets; **PANGU**, Planet and Asteroid Natural Scene Generation Utility; **PSF**, Point Spread Function; **RGB**, Red, Green and Blue; **SfM**, Structure from Motion; **SISPO**, Space Imaging Simulator for Proximity Operations; **SPICE**, Spacecraft Planet Instrument Camera-matrix Events; **SSSB**, Small Solar System Body.

Airbus Defence and Space has developed SurRender, which renders realistic images with a high level of representativeness for space scenes [3]. It uses ray tracing to simulate views of scenes composed of planets, satellites, asteroids and stars, taking into account the illumination conditions and the characteristics of the imaging camera through a user-defined Point Spread Function (PSF). The textures are accessed in a large virtual file, or procedural texture generation can be used. SurRender uses the different models for Bidirectional Reflectance Distribution Function (BRDF), for example Lambertian [4] or [5–7] for the Moon and asteroids, [8] for the Jovian moons. The University of Dundee, UK has developed the Planet and Asteroid Natural Scene Generation Utility (PANGU), which generates realistic, high-quality, synthetic images of planets and asteroids using a custom graphics-processing-unit-based renderer, which includes a parameterisable camera model [9]; it also has a graphical user interface, which makes it more intuitive to use. PANGU implements a Spacecraft Planet Instrument Camera-matrix Events (SPICE) interface, which provides historical and future ephemerides of the Solar System and selected spacecraft. The standard Lambertian diffuse reflectivity model is included as well as Hapke, Oren–Nayar, Blinn–Phong and Cook–Torrance BRDFs. NASA's Navigation and Ancillary Information Facility developed the internationally recognised Spacecraft Planet Instrument Camera-matrix Events (SPICE) tool, which provides the fundamental observation geometry needed to perform photogrammetry, map making and other kinds of planetary science data analysis [10]. It is a numerical tool that provides position and orientation ephemerides of spacecraft and target bodies (including their size and shape), instrument-mounting alignment and field-of-view geometry, reference frame specifications, and underlying time-system conversions; however, it does not have surface-rendering capabilities, and it is limited to shape rendering by implementing the digital shape kernel system (tessellated plate data and digital elevation models). SPICE has the three-dimensional (3-D) visualisation application Cosmographia [11], and has been recently implemented in the toolkit with rendering capabilities for spacecraft orbit visualisation and depictions of observations by probe instruments in Blender [12]. Hapke model also has been used with Blender previously [13].

The comparison between various simulators that are capable of SSSB synthetic image generation is summarised in Fig 1 (camera, orientation and exact light parameters may differ between simulator set-ups). The 25143 Itokawa asteroid model by the [14] was used.

The Space Imaging Simulator for Proximity Operations (SISPO) has been developed to provide a full pipeline from simulated imagery to final data products (e.g., 3-D models); to include photorealistic, physically based rendering; to support automatic surface generation with **procedural displacement textures**; to allow the implementation of spacecraft, instrument and environmental models (e.g., imaging distortions, gas and dust environment, attitude dynamics); and to avoid legacy software. SISPO uses Cycles rendering with the Blender software package, which allows for programming procedures in Python [15]. The preliminary results of 3-D reconstruction and localisation using SISPO, without providing details for simulated imagery and near environment (i.e., gas and dust), were published by [16]. SISPO works on large scales and could simulate a variety of objects at the Solar System scale. It could also be used to generate realistic videos from individual frames, either for visualisation purposes or public outreach.

This article demonstrates the synthetic image simulation capabilities of SISPO, and 3-D reconstruction as a case study showing how such images can be utilised. It also discusses SISPO application to actual space missions, architecture, rendering system and supplementary models.

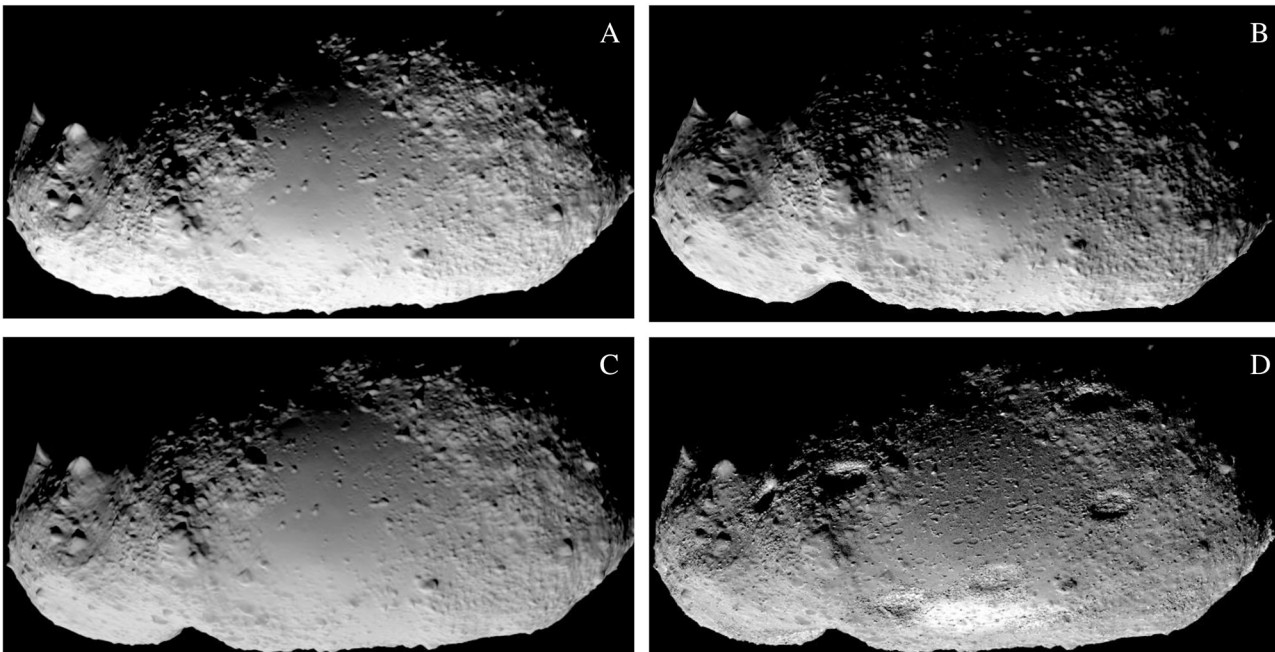

**Fig 1. The comparison of available simulators for space-scene image rendering.** The rendered example for each simulator is asteroid 25143 Itokawa. (A) SurRender image generator by Airbus; it uses backward ray tracing or image generation with Open Graphics Library (OpenGL). (B) PANGU image generator by University of Dundee, UK and ESA; it uses fractal terrain generation using OpenGL. (C) SISPO by the University of Tartu, Estonia and Aalto University, Finland; it uses Blender Cycles physically based path tracer (the same model as above with a simple diffuse shader); (D) SISPO with Blender Cycles physically based path tracer (with procedural displacement with new surface features and reflectance textures for extra detail).

## Application to space mission designs

An increasing number of missions and mission concepts for studying SSSBs require advanced and autonomous operations. For instance, Hera's Asteroid Prospection Explorer daughter-craft (previously called ASPECT) has to navigate autonomously within the proximity of Didymos and its natural satellite [17]; The M-ARGO might be the first-ever standalone nanospace-craft to rendezvous with an asteroid [18]; CASTAway is a mission concept to fly by and explore 10–20 main-belt asteroids and use optical navigation in their proximity [19]; a proposed mission, Castalia, to the main-belt comet 133P/Elst–Pizarro would reveal much that is currently unknown about it and detect water *in situ* in the main belt [20]; Caroline was proposed in 2010 to also fly by a main-belt comet [21]; Titius–Bode is a mission concept to investigate a sequence of asteroids by orbiting its targets for about six months and dispose the Bode lander on the surface [22]; The Martian moons Deimos and Phobos were also targeted by DePhine [23] and Phobos Sample Return [24]. Development and planning of these missions and similar ones require a sophisticated and realistic simulator.

The initial SISPO package was developed to assist in the Multi-Asteroid Touring (MAT) mission, where a fleet of nanospacecraft, propelled by electric sails, flys by a large number of main-belt asteroids [25, 26]. The MAT concept cannot rely on typical deep-space-network-based solutions to operate and navigate the fleet; it requires most of the operations to be performed autonomously. The case study of MAT performing a Didymos fly-by has been evaluated using SISPO; the set of images was generated for the reconstruction and basic localisation for fly-by distances of 33–300 km [16]. The MAT mission was proposed to the European Space

Agency (ESA) announcement of opportunity for "New Science Ideas" [27] and it was not chosen despite reaching the final three [28].

Currently, the SISPO simulator is actively used for Optical Periscopic Imager for Comets (OPIC) development [29], which will be hosted on one of three spacecraft making up the Comet Interceptor mission (https://www.cometinterceptor.space). Comet Interceptor is ESA's first F-class mission (in cooperation with Japan Aerospace Exploration Agency (JAXA)) to fly by either a dynamically new comet approaching the Sun for the first time from the Öpik—Oort cloud, an interstellar object, or a long-period comet as a backup target [30]. OPIC will use automatic image capturing—algorithms will be developed and tested using photorealistic 3-D renderings and a reconstruction pipeline of SISPO using a set of possible encounter velocities and geometries, as well as cometary and camera properties. From a scientific point of view, the simulator will also be used to develop image-prioritisation algorithms, which are required due to the limited data budget, short fly-by timeline and the possibility of the probe being damaged by high-velocity dust impact. The EnVisS coma mapper [31] of the Comet Interceptor mission also uses SISPO for algorithm development.

## General architecture

The general structure of SISPO comprises the following parts:

1. Core features:

   a. Image rendering using Blender Cycles (see Subsection: Rendering in Blender Cycles) or OpenGL (see Subsection: Lightweight OpenGL-based rendering);

   b. Simulation of on-board image processing. Currently the image processing is primarily related to compression; however, inclusion of both cropping and image prioritisation is planned;

   c. 3-D reconstruction (see Section: Usability of images produced for 3D reconstruction).

2. Additional models:

   a. Gas and dust environment (see Subsection: Gas and dust);

   b. Camera distortions (see Subsection: Camera);

   c. Attitude dynamics in the initial stage.

The core functionalities are split into three subpackages. The first subpackage uses Keplerian orbit data for an SSSB and a simplified definition of the encounter geometry so the spacecraft can propagate realistic trajectories using the Orekit library [32] and render an image series of the encounter. The second subpackage provides various algorithms for image compression and decompression. The third subpackage uses images to reconstruct a textured 3-D model using the Structure from Motion (SfM) technique. The three subpackages combined provide a processing pipeline from an initial 3-D model to a reconstructed 3-D model via rendered and compressed images. SISPO is a Python software package that is hosted on a public *GitHub* repository under a *GPL v3.0* licence and is maintained by the authors [33] among other contributors (e.g., authors of this paper). A description of the core functionality and the additional models is provided in Fig 2.

## Rendering system

The most crucial part of SISPO is image synthesis. Two separate rendering modes are implemented: Blender Cycles and OpenGL.

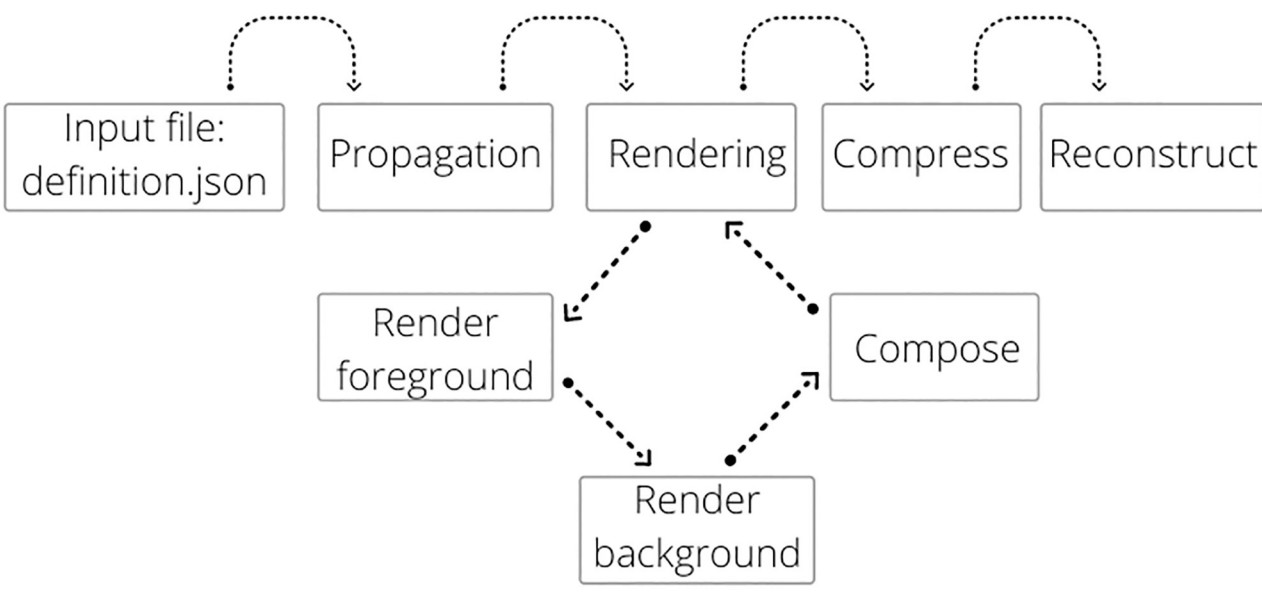

**Fig 2. Program flow of SISPO.**

## Rendering in Blender Cycles

Blender Cycles uses path tracing, which is a type of ray tracing. In classic ray tracing [34], each camera's pixel shoots one or multiple rays, which interact with a surface and then encounters light sources directly or interacts with other surfaces before reaching the light. For realistic reproduction, various optimisations can be used. The ray-surface and ray-volume interactions can be modelled by shaders, which model or approximate the interacting objects' properties.

The Cycles rendering engine supports various shaders, the most relevant of those for this paper being: (i) diffuse bidirectional scattering distribution function, which provides access to Lambertian and Oren–Nayal shaders based on surface roughness; (ii) emission shader that allows surfaces or volumes to emit light; (iii) subsurface scattering that supports cubic, Gaussian and Christensen–Burley models; (iii) glossy shader that supports Sharp, Beckmann, *GGX*, Asikhmin–Shirley and multi-scatter *GGX* models; (iv) volume scattering shader that allows simulating light scattering in volumes [15]. These shaders can be combined with both procedurally generated or pre-existing texture maps to change their parameters and mix various shaders on and within the scene models. Cycles also supports Open Shading Language, which allows the use of arbitrarily defined shaders.

Path tracing is, in some way, an improvement on ray tracing; it produces multiple rays from the same pixel in random directions, then each ray keeps bouncing (without producing new rays) until it reaches the light source or user-defined bounce limit. In Blender Cycles, one can limit the following bounce limits (ideally it should be infinite; however, typically, a limited amount is sufficient): total, diffuse, glossy, transparency, transmission and volume scattering. Then, the amount of light per pixel is calculated for each ray along with surface colour properties; afterwards, the value is averaged and assigned to that specific pixel. Moreover, Blender has a branched path tracer, which can be used for volumetric scattering (see Subsection: Case 2: volumetric particle effects). The branched path tracer is similar to path tracing, but at the first ray interaction it will split the path for different surface components, and for shading, it will take all lighting parameters into account instead of just one [15]. The branched path tracer can also be useful for solving light-related problems such as caustics.

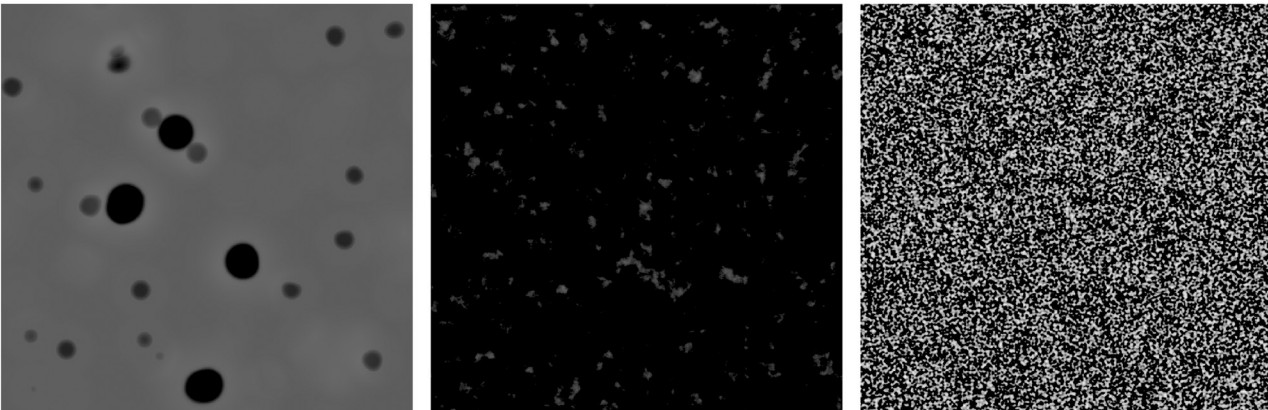

**Fig 3. Procedurally generated height maps.** For craters (left), rocks (centre) and sand (right).

All the surface features can be generated procedurally (e.g., by applying mathematical equations instead of externally-captured image texture) inside the Blender software [35]. The SISPO surface model includes sand flats, rock formations and craters. The size of these features, their number and their distribution can be adjusted by modifying the corresponding parameters.

The elevation details are simulated by using height maps for craters, rocks and sand inside the rendering engine (Fig 3). These consist of textures converted into surface displacement distances and are used to displace parts of the model mesh and provide surface normal changes.

Height maps can be mixed together to simulate different types of terrestrial bodies (see Section: Simulations of realistic asteroid imagery). Each map's weight in the mix can be modified, or even obliterated, and they can be made to affect only specific areas of the mesh by using masks. The shader adds a texture whose albedo corresponds to the albedo measured in real asteroids in addition to the procedural elevation. The final procedural shader example is shown in Fig 4.

### Lightweight OpenGL-based rendering

There is an option to use a custom OpenGL-based renderer instead of Cycles and Blender. It was added because it is desirable to generate images fast in certain situations even though some fidelity (e.g., softer shadows or procedurally generated details) might be lost. For instance, during the development of image-processing algorithms, it is useful to generate large datasets for training data in a relatively short time. Validation data can be generated with Cycles to give better confidence in the characterised algorithm performance. Fast image generation also enables a closed-loop simulation of a guidance, navigation and control system that depends on the navigation camera input.

Three different BRDFs have been implemented as OpenGL shaders: Lambertian [4], Lunar-Lambertian [36] and Hapke [37]. Textures are supported and correspond to the single scattering albedo of the corresponding shape model region.

Efficient shadowing is achieved by shadow mapping [38]. Shadow mapping works by first rendering the scene orthographically from the direction of the Sun. The resulting depth buffer contents are imported to the actual rendering pass as a texture. The object vertices are again projected orthographically towards the Sun; the corresponding fragment is considered to be in the shadow if the depth of that projection is greater what is in the shadow texture.

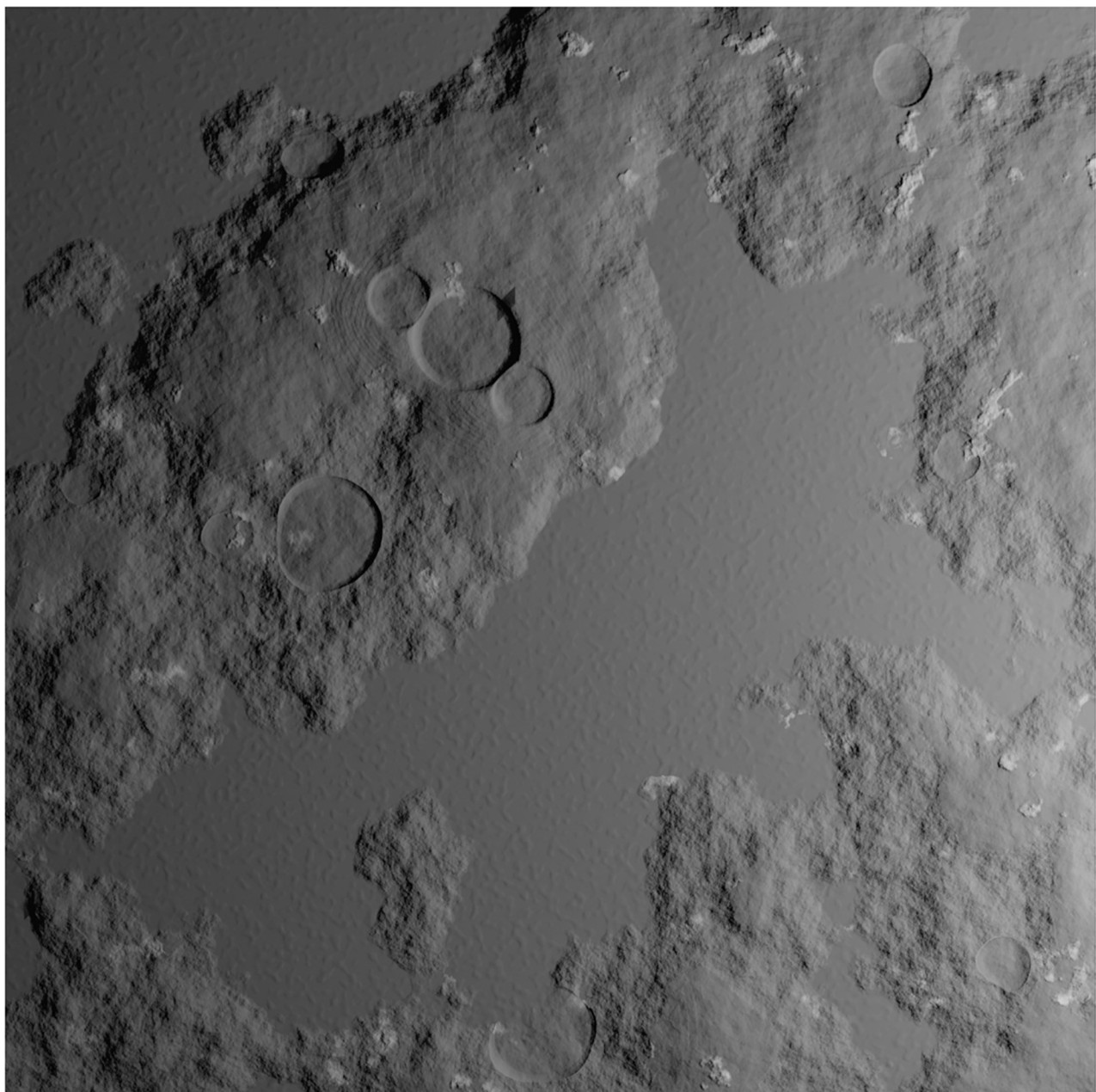

**Fig 4. Final procedural shader applied to a subdivided plane.**

The output of the OpenGL render is a floating-point value giving the irradiance $[W \cdot m^{-2}]$ incident on each theoretical pixel. This "irradiance image" can be further passed to the default camera model used by SISPO, which then applies appropriate imaging effects to get the final synthetic image. There is currently no procedural shape or texture generation due to the simplified nature of the OpenGL renderer. Dust and gas coma rendering is not fully integrated with the OpenGL renderer. Instead, emission-based rendering of a previously generated coma is done separately in Python after OpenGL rendering (see Subsection: Case 2: volumetric

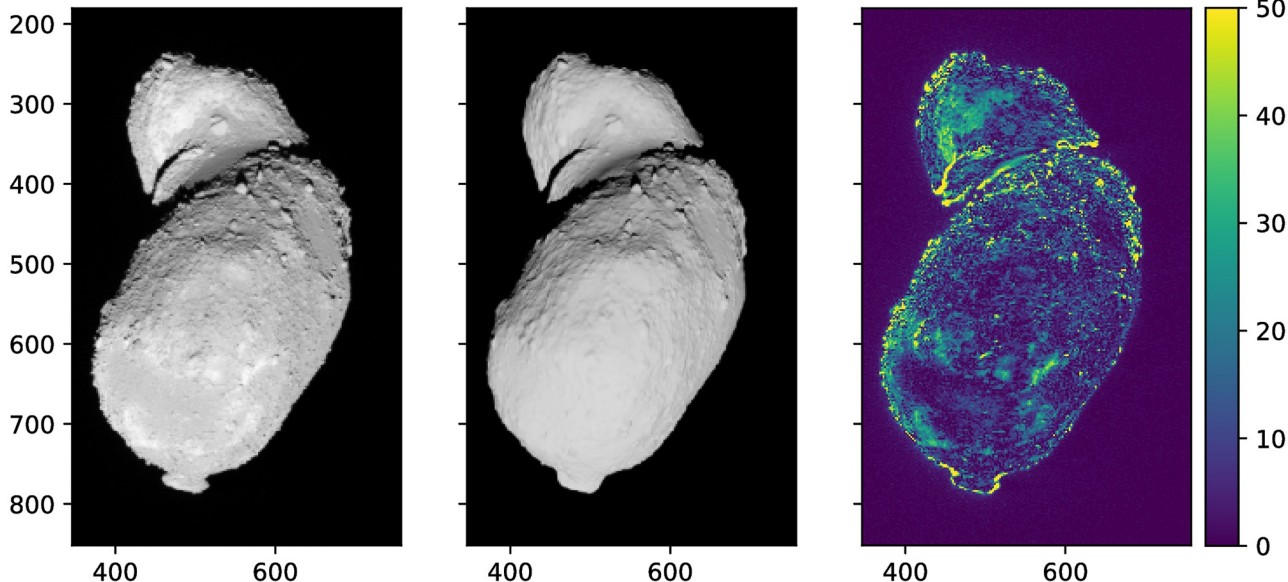

**Fig 5. The pixel-by-pixel comparison of the original AMICA image on the left and generated OpenGL image in the middle.** The scale indicates percent error. The AMICA image is published under Public Domain 1.0.

particle effects), and it does not take into account shadows cast by objects in the scene. One fundamental limitation to OpenGL-based rendering is that it does not support extended light sources. Thus, for instance, earthshine in low Earth orbit—or in the case of a binary asteroid, the reflected light from the primary body to the secondary body—cannot be reasonably modelled. As the light from reflecting surfaces is not considered, the shadow of rocks, cliffs and other geological formations will be darker than in reality.

The OpenGL rendering engine was used for a preliminary guidance, navigation and control analysis done for the preliminary design review of the now-defunct Asteroid Prospection Explorer project. For this purpose, the rendering engine was interfaced from a Simulink model designed to handle the system dynamics. The same model then forwarded the generated images to the visual navigation algorithms for position estimates. The generation of one image took around 0.7–1.0 seconds. The simulation time for one mission week was around 12 hours on a laptop with an Intel i7–7700HQ (2.8 GHz) processor, with most of the processing time spent on executing the visual navigation algorithms.

Fig 5 demonstrates the comparison between the image taken by the AMICA instrument of the Hayabusa spacecraft and rendered image using OpenGL. The main rendering discrepancies are induced by the inaccuracy of the 3D model [14] and local variations of albedo and roughness.

## Supplementary SISPO models

### Gas and dust

In the inner Solar System, the solar radiation heats nuclei of comets, which causes them to release gas together with dust. This gas and dust surround a comet forming a coma, which is blown by the solar wind resulting in ion and dust tails that can extend over 100 million kilometres and might be visible from Earth. Sometimes the comet releases strong outbursts of these gasses and dust to produce distinctive coma-derived features called jets [39]. The gas and

dust environment is visible to the camera instrumentation, and hence, in order to provide realistic space-scene simulation, the visible effects should be included in modelling.

The dust and gas environment causes noise that can affect the 3-D shape reconstruction. SISPO offers models for comae, jets and tails with various details. The problem can be divided into two parts: modelling the environment and rendering it. The modelling parts can contain, for instance, the description of the environment, such as jets represented as geometric cones from the surface of the comet, mathematical models from [40], or gas and dust simulations such as those from [41, 42].

The rendering problem comes down to the level of desired details and accuracy, but on the other hand, to reasonable computing time. In reality, the volume is a mixture of gas and dust, which are composed of different particle sizes with different densities, and this produces various scattering characteristics [43]. Realistic and accurate representation of the effects mentioned above in the rendering pipeline, which is not specialised in volume scattering, is a challenge; a visually realistic approximation should be sought. In the current version of SISPO, volume-scattering effects from the coma and jets are computed using a simple volume-scattering shader from Cycle that uses the voxel presentation of the comet's surroundings as an input for the density.

## Camera

The Optical Aberrations for Still Images Simulator (OASIS) provides simulation tools for optical aberrations that are usually not implemented in popular 3-D software (e.g., Blender). Since it uses two-dimensional images as input, motion blur effects, which require spatial awareness of a scene, are not modelled. Optical Aberrations for Still Images Simulator (OASIS) is used in a complementary manner with SISPO to further enhance rendered two-dimensional output images.

Tangential and sagittal astigmatism, as well as an internal and external comatic aberration, are modelled with distinct PSFs, which vary with the field height and orientation of the sensor centre—image point vector. By default, aberration intensity increases with the field height. However, a custom lens file can be provided to model any desired lens array.

Lateral chromatic aberration is modelled by rescaling individual colour channels and simulating wavelength-dependent refraction of light rays. While wavelengths are continuous in the real world, the digital Red, Green and Blue (RGB) triplet only distinguishes between three discrete primitive colours, which introduces sharp edges at the boundaries of colour separation. These can be avoided by blending chromatic aberration with tangential astigmatism (more information with an example can be found in Section 5.7 in [44]).

Dark-current noise is currently drawn from a folded normal distribution with a zero mean and user-defined standard deviation. Readout noise is modelled by the addition and subtraction of random values at the subpixel level, governed by a Gaussian custom standard deviation process. The projection of light rays with random origin positions generates realistic shot noise and follows the user-defined average sample size per pixel.

Lens distortion is simulated with a sophisticated Brown–Conrady model [45], that corrects for both radial and tangential distortion. It is implemented in the computer-vision library OpenCV [46] and supports up to six radial distortion coefficients—$k_1$ to $k_6$—and two tangential distortion coefficients—$p_1$ and $p_2$.

Monochrome sensors are modelled by averaging an RGB colour triplet of a virtual light ray. Additionally, a weighted-average model can be selected that reflects the indeed perceived luminosity. The generation of dark current and readout noise is adjusted accordingly to maintain a specified standard deviation on monochromatic detectors. The simulation of sagittal

astigmatism, coma, chromatic aberration, shot noise and readout noise is demonstrated in more detail in [44] (Section 5.7).

Currently, OASIS is limited to simulating one type of PSF at a time, as a realistic convolution of multiple PSFs is not yet provided, with the exception of small aberrations. Also, the generation of dark noise, which should follow a Poisson distribution, is subject to change once physical units are implemented for photon flux and for the conversion between light-ray energy and digital-sensor value.

## Orbital simulation

The trajectory simulation within SISPO is handled by the Orekit library [32]. A simple Keplerian orbit propagator is used to propagate both the SSSB and the spacecraft. Additionally, it is possible to rotate the SSSB around a single axis at a constant spin rate.

The implemented Orekit Python bindings run a virtual machine to execute the underlying Java code. During propagation, Orekit determines state information of the SSSB and the spacecraft for each sample along the trajectory. The state information includes the date, position and the rotation angles of the SSSB.

In SISPO, the spacecraft trajectory is normally defined by its Keplerian elements, but the user does not explicitly enter these elements. Instead, the elements are calculated from the target body's Keplerian elements and the expected encounter geometry relative to the SSSB at the closest approach. The parameters presented in Table 1 are used to calculate the state vector of the spacecraft at the encounter, which defines the spacecraft trajectory. The *sssb_state* is calculated based on the SSSB input data and the encounter data.

The propagation process is defined by the duration of a fly-by, the number of frames to be rendered, *timesampler* mode and a *slowmotion* factor as presented in Table 2. The *timesampler* mode determines whether the steps are distributed linearly in time (mode 1, default) or whether an exponential model (mode 2) is used, which increases the number of frames around the encounter. The number of additional samples can be controlled with the *slowmotion* factor.

**Table 1. Parameters defining the encounter state of the spacecraft in SISPO.** The first five parameters are required as input.

| Parameter | Unit | Type | Description |
|---|---|---|---|
| encounter_distance | m | *float* | Minimum distance between SSSB and spacecraft. |
| with_terminator | – | *bool* | Determines whether the terminator is visible at the closest approach. |
| with_sunnyside | – | *bool* | Determines whether the spacecraft passes the SSSB on the Sun-facing side or the side facing away from the Sun. |
| relative_velocity | m s$^{-1}$ | *float* | Relative velocity of the spacecraft to the SSSB at the encounter. |
| encounter_date | – | *dict* | Date of the closest approach of the spacecraft and SSSB. The type is a Python dictionary with an *int* for year, month, day, hour, minute, and *float* for second. |
| sssb_state | m, m s$^{-1}$ | *tuple* | SSSB state vector containing three position and three velocity components at the encounter. The spacecraft encounter state is calculated relative to this state vector. The SSSB state is not required as input since it is calculated based on the SSSB trajectory and encounter date. |

**Table 2. Input parameters that define the propagation step in SISPO.**

| Parameter | Unit | Type | Description |
|---|---|---|---|
| duration | s | *float* | Total length of the simulation. The encounter date is reached after half of the duration. |
| frame | – | *int* | Number of frames (samples) taken during the encounter. |
| timesampler_mode | – | *int* | Mode 1 is linear and mode 2 is exponential sampling. |
| slowmotion_factor | – | *float* | Determines how many more state samples are taken around the encounter. Only applies if timesampler_mode is exponential. |

Mode 2 is especially helpful when simulating a long fly-by, since, when the spacecraft is far from the SSSB nucleus, only minor changes are visible in rendered images.

## Attitude dynamics

Along with the orbital simulation, a simplified attitude dynamics portion is already built into the Orekit framework, also accessible via the SISPO library. Attitude information in Orekit is handled essentially as another frame transformation. It contains the rotation from the reference frame to the satellite frame, and then the angular velocity (i.e., spin) and angular acceleration (i.e., rotation acceleration) of the spacecraft in its frame. This makes Orekit's *spacecraft_state* a great construct to hold attitude-related and orbital data, which can then successfully be propagated in time. The Orekit library also allows SISPO to build user-defined "attitude law". This attitude law can be selected from several pre-existing and common attitude modes (e.g., pointing, spin stabilised) or recompiled entirely in a way necessary for the mission simulation.

It is, however, essential to note that the Orekit library mainly focuses on orbital mechanics and propagation. The attitude model, and any attitude laws it follows, presumes the existence of a "perfect attitude control system" without necessarily considering the physical limitations or perturbations of the satellite attitude by external forces. The Orekit library lacks the necessary constructs to convey and utilise the moments of inertia of the satellite. Since this information is crucial for the inclusion of external forces and their effect on the spacecraft's attitude or other celestial bodies, this will be implemented as a secondary layer into the current framework; this is explained in Section: Discussion and future work.

## Simulations of realistic asteroid imagery

This section demonstrates five different use cases of the SISPO software.

### Case 1: Asteroid 25143 Itokawa

Fig 6 demonstrates the comparison between the image taken by the AMICA instrument of the Hayabusa spacecraft and rendered image using Cycles on the plain model using a simple diffuse shader. The main discrepancies in the rendering are due to the inaccuracy of the 3D model [14]. The higher error spots are caused mainly by the local variations of albedo and roughness. The biggest advantage of the procedural texturing, which introduces new surface features, is demonstrated in Fig 7. The fragmented zoomed view in Fig 8 demonstrates how surface features are added procedurally and the level of details that can be achieved in comparison with the plain mesh.

Fig 7 shows a comparison between a plain mesh of the asteroid 25143 Itokawa and a mesh recreation using the procedural shader in Blender. The initial mesh is a 3-D reconstruction of the actual asteroid as described in [47]. The shader then adds procedural surface features on top.

A small area of the model mesh was rendered using OpenGL and Cycles to show the ultimate advantage of procedural texturing. Both renders are shown in Fig 8, and the one with procedural texturing indicates the capability to generate almost arbitrary level of detail. This feature would be required, for example, to train probes that can land on the surface.

### Case 2: Volumetric particle effects

Coma and dust jets are produced by particles emanating from the surface, and these are used to produce a volumetric density distribution. For the preliminary proof of concept, the jets

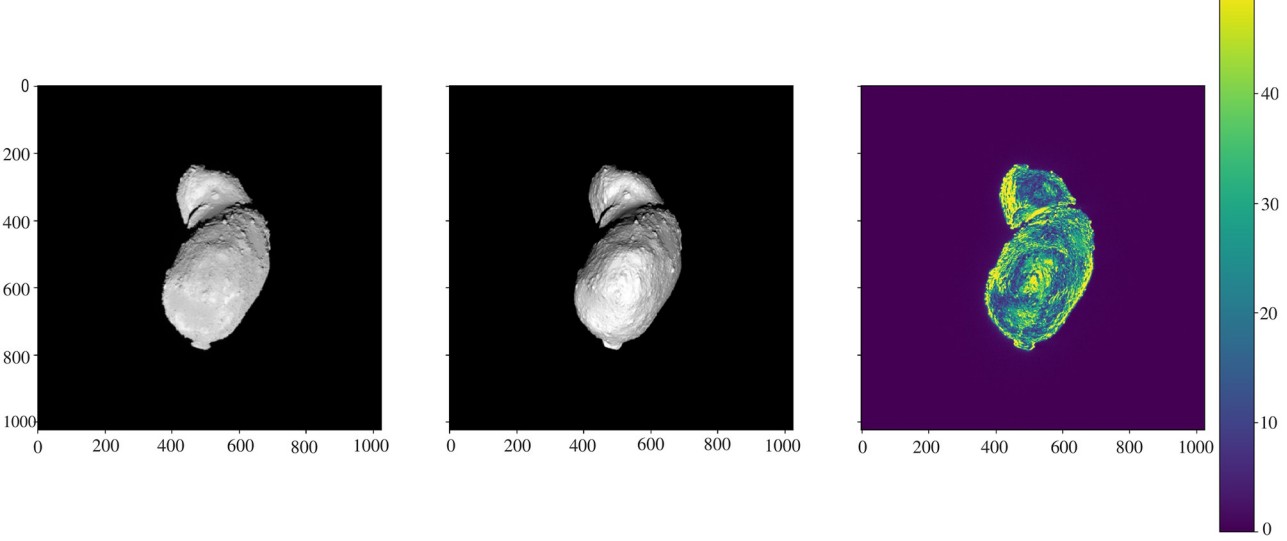

**Fig 6. The pixel-by-pixel comparison of the original AMICA image on the left and generated Cycles image in the middle.** The scale indicates percent error. The AMICA image is published under Public Domain 1.0.

were modelled using simplified versions of the gas and dust models from [41, 42]. First, the gas source strengths and gas velocities are generated for each face of the mesh from the noise function. The gas source data are then used as input for the gas model [41], which computes the gas field densities and velocities around the comet. This data is then used in a particle simulation following [42] from which the particle position data is further processed into a three-dimensional number-density map around the comet, which is encoded to a single OpenEXR file [48]. Before the render, the number-density map is smoothed with tricubic interpolation [49]. It can then be loaded by Cycles and rendered using either volumetric scattering or emission (volumetric emission, although less accurate, is orders of magnitude faster).

**Volumetric particle effects on the comet 67P/C–G.** In Fig 9 (the coma might appear differently to the article's viewer depending on the monitor or the print quality), the dust environment capabilities of SISPO are presented by simulating and rendering images of the comet 67P/C–G. This image is then compared to the actual image, and with an OpenGL rendered

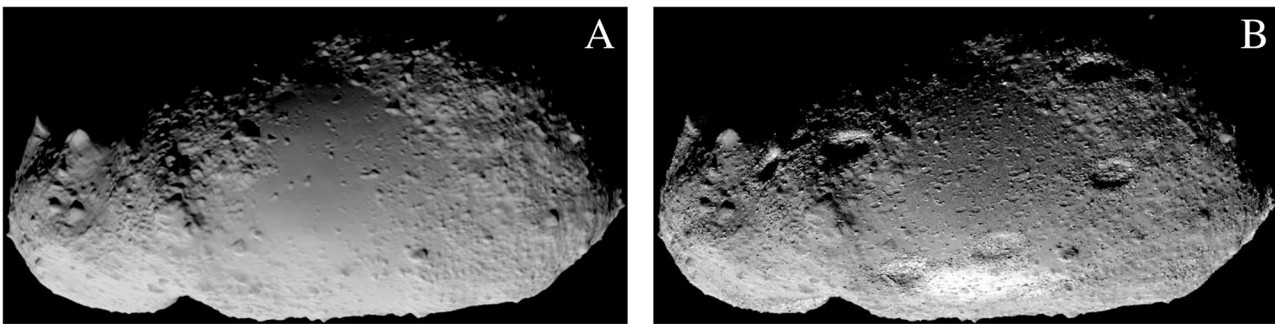

**Fig 7. Surface mesh of asteroid 25143 Itokawa.** Created by the AMICA imaging team [14] and rendered by authors in Blender Cycles. (A) plain mesh and (B) smooth mesh with procedural shader.

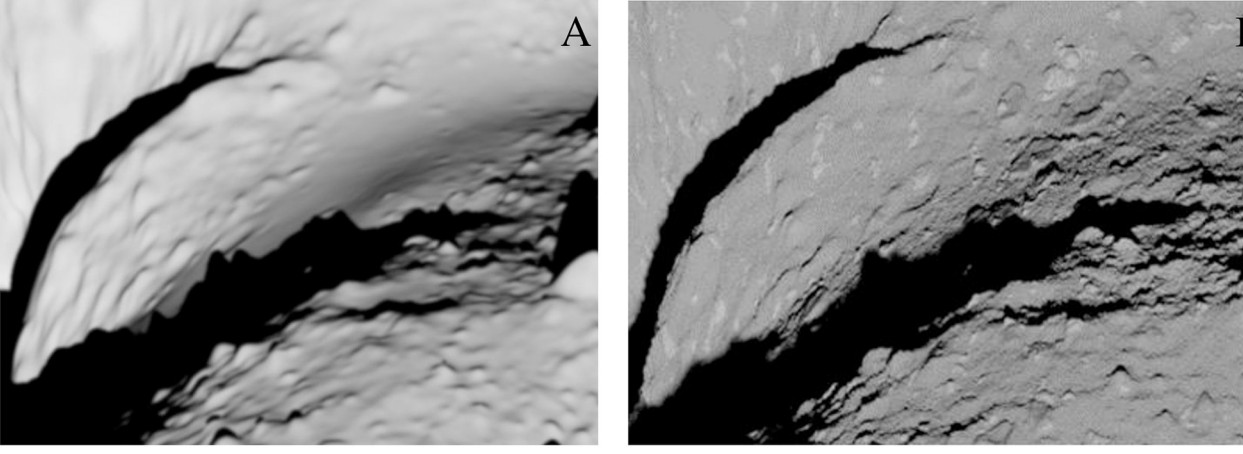

**Fig 8. Close view of the 25143 Itokawa surface.** Zoomed-in region. (A) OpenGL render and (B) Cycles render.

image. All the shader features (rocks, craters and sand flats) can be used simultaneously for complex SSSBs if needed. The masks that separate the "sand flats" from the rest of the surface features are created with procedural noise, but they could be painted by hand if necessary.

## Case 3: Larger bodies

The reproduction of larger terrestrial bodies in SISPO is demonstrated in Fig 10, which is based on the narrow-angle-camera digital terrain model of the Apollo 15 landing site and operations area [50], obtained by Lunar Reconnaissance Orbiter Narrow Angle Camera (http://wms.lroc.asu.edu/lroc/view_rdr_product/NAC_DTM_APOLLO15_M111571816_50CM, accessed 28.01.2021). The original terrain model has a 2 m resolution for elevation maps and 0.5 m resolution for orthographic photos, which is not enough to produce images from the surface that would be useful for navigation testing. This case demonstrates the image scalability produced in SISPO. The lunar rendering was made by implementing a digital

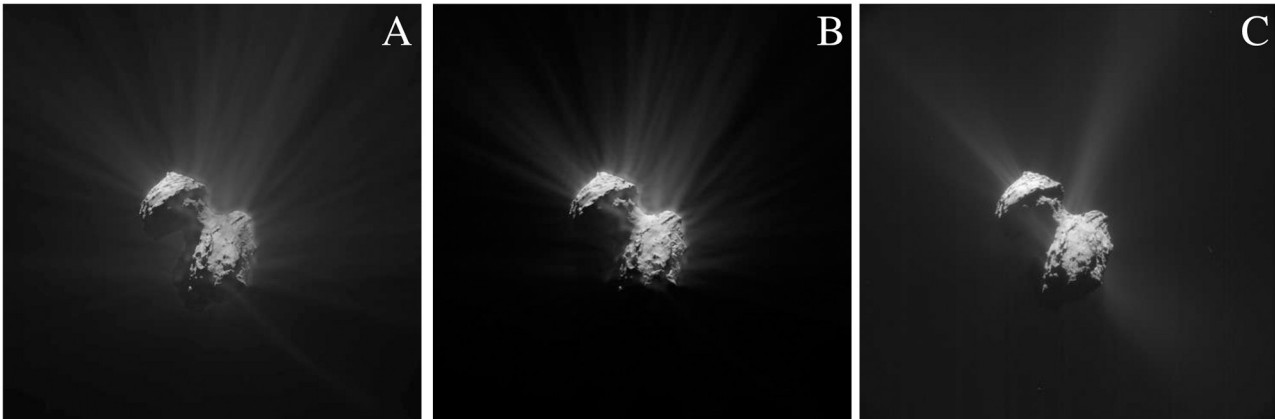

**Fig 9. Comet 67P/C-G with coma.** (A) OpenGL, (B) Cycles and (C) Real image. The brightness was enhanced at the post-processing stage in order to better visualise coma.

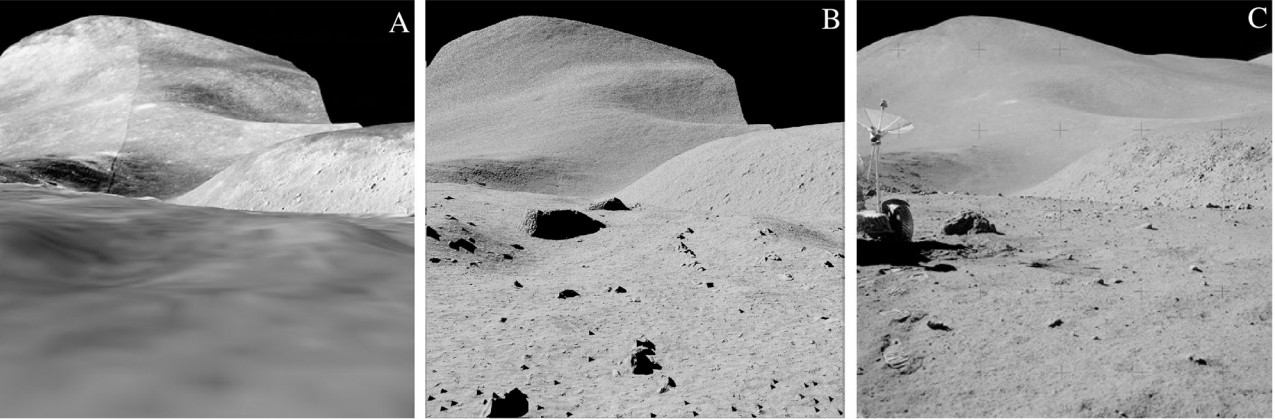

**Fig 10. Application of procedural texturing on the scale of a larger airless body.** (A) using a digital terrain model and applying orthographic photos as a texture. (B) using a digital terrain model and procedural terrain generation with Blender Cycles on the same digital terrain model. (C) image taken during the Apollo 15 mission from roughly the same location.

terrain model and procedural texturing over the whole $5.1 \times 28.6$ km$^2$ area; the farthest point visible on the render is roughly 11 km away from the camera.

## Case 4: Subsurface exploration

Lava caves, the isolated underground environments, exist on Moon [51] and Mars [52]. These caves have been studied by remote sensing and have not been explored by dedicated missions, although some were proposed and developed, such as Moon Diver [53] and rock climber Lemur [54]. Lava tubes have been morphologically related to ones formed on Earth in volcanic rock by a volcanic eruption [55]. SISPO could be utilised for synthesising versatile sets of lava-caves images for (i) developing navigation algorithms and sampling spots detection by image processing (e.g., biological mats or their preserved remains on the geological substrate) or (ii) mapping the caves by photogrammetry for potential human settlements. An example of SISPO cave rendering is demonstrated in Fig 11.

## Case 5: Fly-by of a spacecraft

The SISPO environment is suitable for spacecraft fly-by rendering. This can be useful for multi-spacecraft missions, in-orbit servicing and fleets. The Cycles rendering of the spacecraft fly-by is demonstrated in Fig 12. The model uses a *principled bidirectional scattering distribution function* shader including multiple layers to create spacecraft materials. The set of produced images can be used for developing and training formation-flying proximity algorithms demanded by attitude control and orbit determination subsystem.

## Usability of images produced for 3D reconstruction

A set of synthetically generated images can be used for photogrammetry-based 3-D surface reconstruction within the SISPO environment. The steps executed in the reconstruction pipeline are described in [16]. The reconstruction uses two libraries:

1. Open Multiple View Geometry (OpenMVG) C++ library [56], which creates a sparse point cloud based on two different SfM techniques:

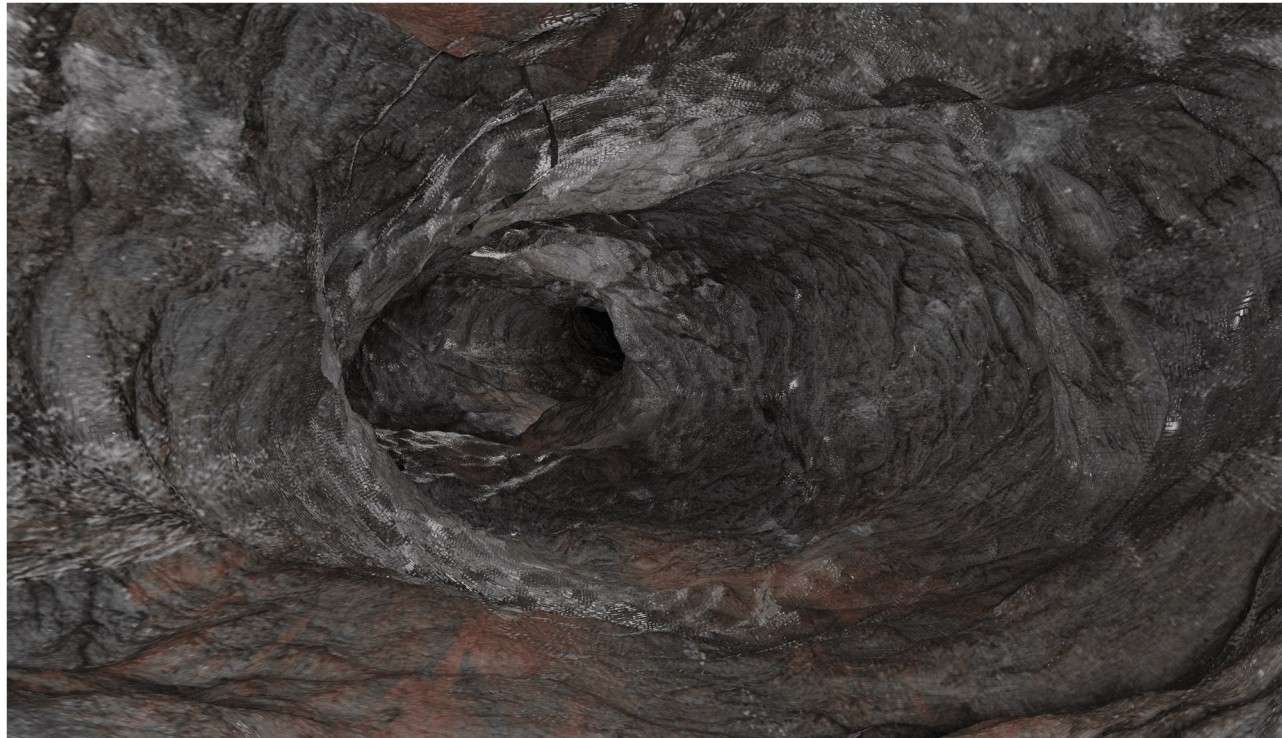

**Fig 11. An example of a cave rendering.** Colour, specularity and normal maps were applied. Colour maps are used from the walls of the Hillingdur and Blámi lava caves in Iceland obtained by Iakubivskyi. White spots indicate possible biological mats or secondary mineral precipitation; the red areas simulate iron oxide.

 a. Global SfM [57].

 b. Incremental SfM [58], which is much more suitable for SISPO applications [16].

2. Open Multiple View Stereo Reconstruction (OpenMVS), which uses input from OpenMVG, creates a dense point cloud, a faceted surface (mesh) or a set of planes [59].

## Case 1: SSSB fly-by

A set of 25 images was generated in SISPO and then used to reconstruct the 3-D model using the pipeline. The images were generated using a pinhole camera (i.e., no optical aberrations,

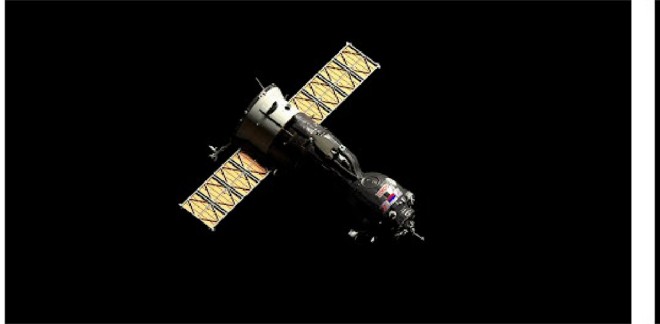
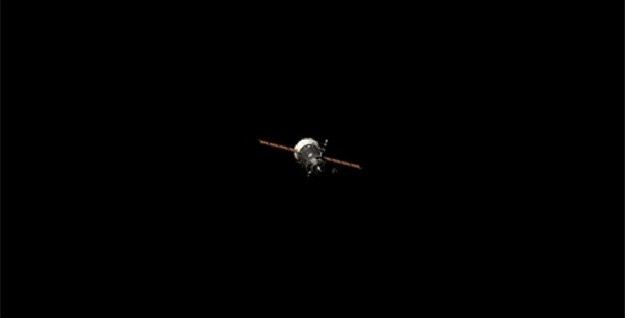

**Fig 12. Rendering of a spacecraft fly-by.**

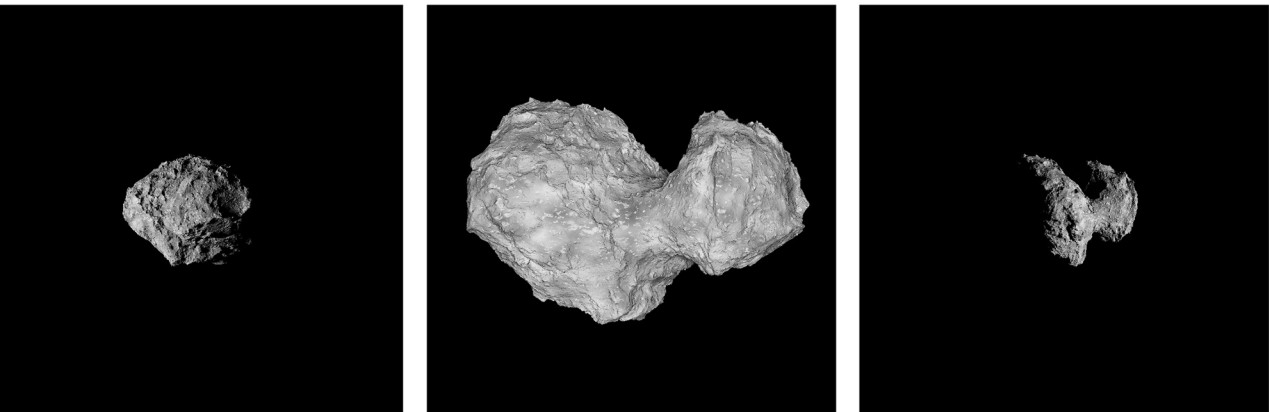

**Fig 13. Three out of 25 frames used for the reconstruction.** Images rendered in SISPO. Furthest approaching frame, closest approach and furthest leaving frame.

which would decrease the reconstructed accuracy; however, they can be implemented as discussed in Subsection: Camera). The two furthest points and the closest approach are shown in Fig 13.

The result of reconstruction is shown in Fig 14, which visualises vertices and the triangular mesh via the inclusion of normals and applied texture in the reconstructed model.

As a result, two 3-D models are obtained: the input model for SISPO and the reconstructed model from images. Only the visible part of the comet was obtained because of the nature of

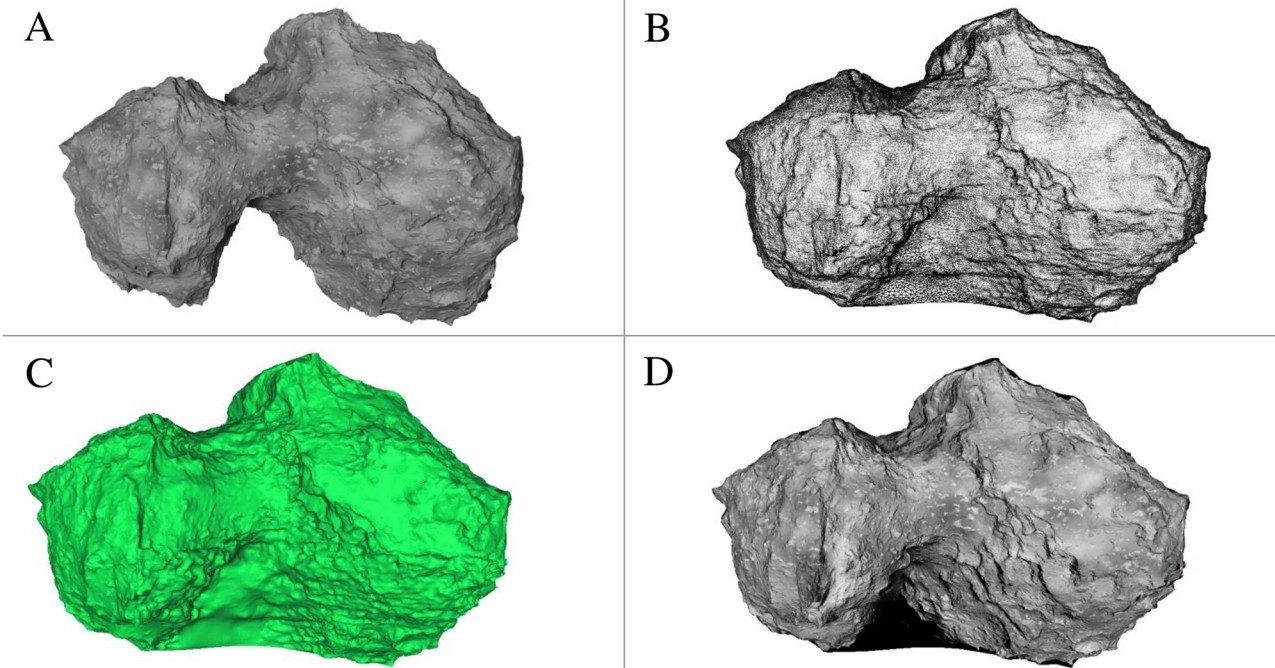

**Fig 14. Final reconstruction of the visible part of the target without post-production.** (A) Synthetically generated image (input). (B) Vertices of reconstruction. (C) Mesh with normals. (D) Triangular mesh with texture.

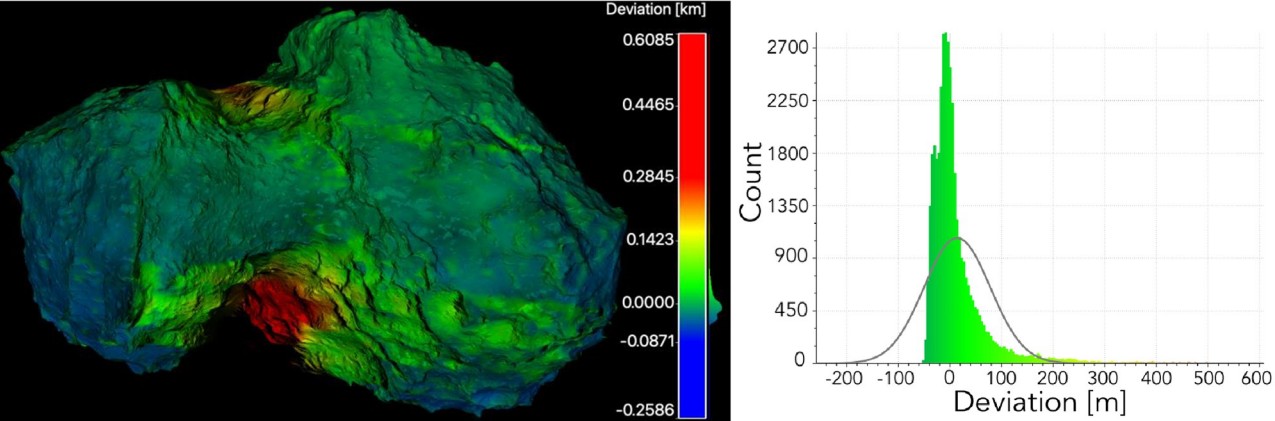

**Fig 15. Deviation analysis of the reconstructed partial mesh and input 3-D model for image synthesis.** Visualisation of deviation (left); errors and Gaussian distribution (right).

the fly-by (i.e., during the fly-by only one illuminated part of the target is visible). The reconstructed model was compared with the input 3-D model externally using the *CloudCompare* software [60]. Generally, the reconstructed model is close to the input model, except for the edges (e.g., the black area in Fig 15), where the error deviation was very high (up to 600 m). These highly deviated parts could be removed in the point cloud or mesh post-processing, but have not been done in this analysis. The Gaussian distribution mean error is 14.7 m and the standard deviation is 63.9 m. The visualisation of deviation and the error analysis are shown in Fig 15.

## Case 2: SSSB orbiting

During SSSB orbiting, it is possible to observe every side of the target because as the target spins, each part will be illuminated at a certain point. Three different input-image views (out of 53 used in total) and a fully reconstructed 3-D model are shown in Fig 16. Fifty-three images were generated from different angles using a pinhole camera.

The input and reconstructed 3-D model were compared using the same method as in the fly-by case. Because of the larger set of observational angles, the deviation is decreased in the orbiting case and reaches 60–89.7 m at some crater spots, but most of the surface is aligned with the input model as shown in Fig 17. The Gaussian mean error distribution is 1.2 m and the standard deviation is 6.6 m. The accuracy can be increased by using a more significant set of images and higher resolution.

## Discussion and future work

SISPO is a sophisticated tool for terrestrial-cosmic-scenery rendering. Its most significant advantage is the physically based, high-quality, realistic renderings, which can reproduce submillimeter surface resolution and beyond for SSSBs and other terrestrial bodies thanks to micropolygon procedural texturing and Blender's Cycles path-tracing rendering engine. The produced renderings are mature for deep-space mission design and algorithm development for semi-autonomous operations, visual navigation, localisation and image processing. There are a few more functions that are considered for future implementation:

- Attitude dynamics auxiliary package (see below);

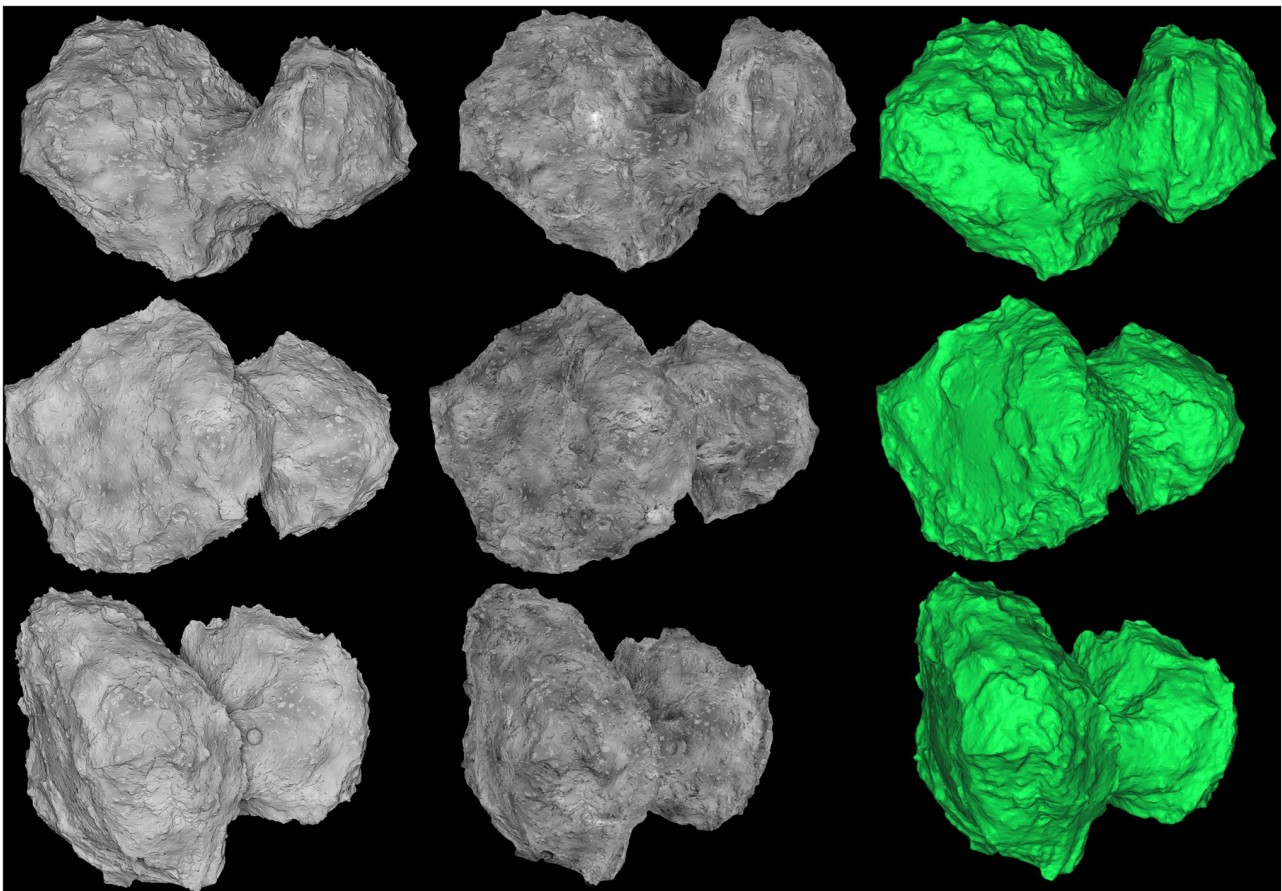

**Fig 16. Full reconstruction of the comet during orbiting.** The first column shows actual synthetic images, the second column is the textured view of the 3-D reconstruction, and the third column shows the meshed view of the reconstruction.

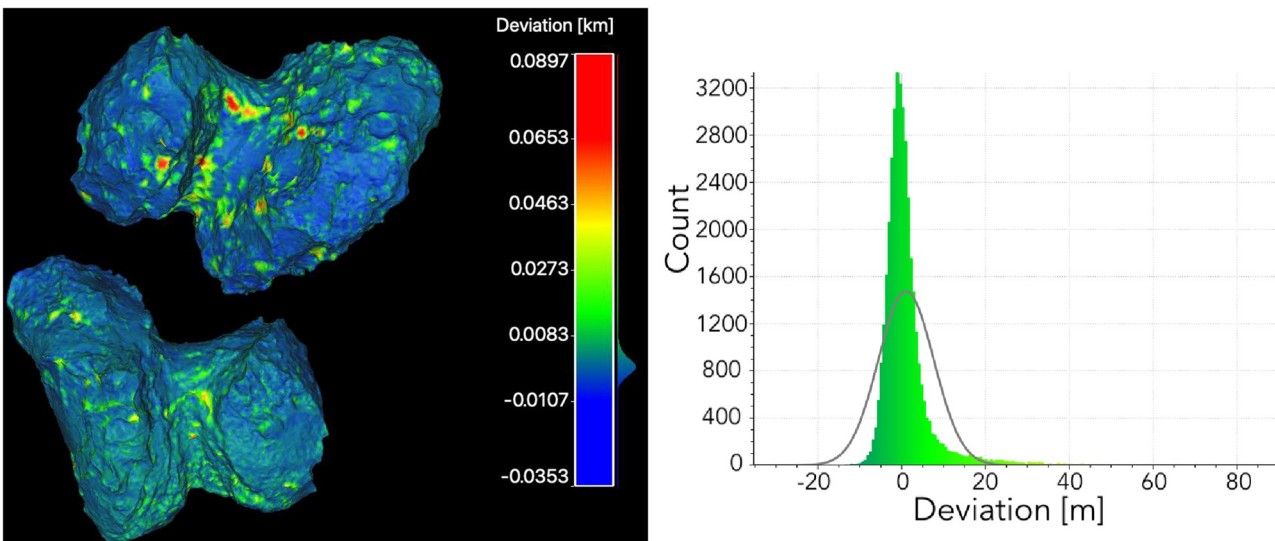

**Fig 17. Deviation analysis of the entire reconstructed mesh and input 3-D model for image synthesis.** Visualisation of deviation (left); errors and Gaussian distribution (right).

- Image compression algorithms for efficient data storage and transmission evaluation, which was preliminarily assessed by [61];

- Reconstruction of various scenery;

- Algorithms for spacecraft photogrammetry-based localisation;

- Capability to simulate measurements of the target spectral reflectance in certain wavelength channels;

- Improved shader and user-defined parameterisation of Cycles in order to minimise the approximation;

- Solar System ephemeris integration for historical and upcoming events. This can include both terrestrial bodies and spacecraft via the possible adaptation of Spacecraft Planet Instrument Camera-matrix Events (SPICE) data.

### Attitude dynamics auxiliary package

The SISPO simulation environment will include optional packages for calculating the effects of external forces such as dust particle impacts, atmospheric drag, gravity gradient and various other effects necessary to account for. Each of these functionalities will be built as an add-on to the current Orekit-based attitude framework. This is done by adding an optional extra calculation step between the propagations of the spacecraft state through time by Orekit to account for the perturbations. With the addition of the second layer on top of Orekit orbital simulations, for accurately simulating the attitude dynamics of the spacecraft or a celestial body, the SISPO environment will have the necessary groundwork for later expansions into different active and passive attitude control systems present in any future mission simulations.

### GitHub algorithm repository

The open-source algorithm is stored in the public GitHub repository (https://github.com/SISPO-developers), and everyone is welcome to use and modify it. The algorithm is updated, and new functionalities are managed and added by authors. The instruction and comments can also be found in the repository. It contains following subrepositories: main SISPO (https://github.com/SISPO-developers/sispo), dust and gas environment generator for SSSBs (https://github.com/SISPO-developers/ComaCreator), comatic aberration and astigmatism simulator (https://github.com/SISPO-developers/OASIS) and docker image (https://github.com/SISPO-developers/sispo_docker) for fast SISPO deployment. Additional information about the algorithm can be found in [44, 61].

### Conclusions

In this paper, the Space Imaging Simulator for Proximity Operations (SISPO) architecture and capabilities were described and several features demonstrated by case simulations. The tool generates physically based, photorealistic images from an input 3-D model using Blender's Cycles path-tracing rendering engine. For regolith surface reconstruction it uses procedural texturing. SISPO has supplementary models for optical aberrations as well as for gas and dust of small bodies. The description and usage of these models were discussed in this paper. Other use cases have demonstrated renderings of the Moon and a spacecraft. The set of produced images can be implemented for 3-D surface reconstruction. The tool is open access and currently requires basic programming skills to be used [33]. The level of detail currently produced

by SISPO is suitable for the design of advanced deep-space missions, the simulation of large sets of scenarios, and the development and validation of algorithms for (semi-)autonomous operations, vision-based navigation, localisation and image processing. SISPO has already supported the development of deep-space mission concepts and is currently being used for the ESA–JAXA Comet Interceptor mission (more details in Section: Application to space mission designs).

Some further improvements have been considered and are listed in Section: Discussion and future work. Other teams are welcome to use SISPO, implement new functions and contact the authors of this paper.

## Supporting information

**S1 Script. Image comparison.** The file contains python algorithm used to compare images in this manuscript.
(ZIP)

## Acknowledgments

We thank Mattias Malmer for allowing us to use his shape model of the comet 67P/Churyumov–Gerasimenko. Thanks to Hayabusa's AMICA imaging team for publishing the shape model of 25143 Itokawa. We would like to express our gratitude to Airbus Defence and Space for developing and providing the SurRender software; to Jérémy Lebreton and Christine Pelsener-Scamaroni in particular for providing and accommodating changes to the licence agreement. Thanks to the University of Dundee and Martin Dunstan for developing and providing PANGU software and helping to set it up. A special mention goes to the Blender community, thanks to whom the open-source software exists and improves. Thanks to Tomas Kohout (University of Helsinki) for helping to arrange Timo Väisänen's civilian service (a.k.a. semi postdoc) at Aalto University.

## Author Contributions

**Conceptualization:** Mihkel Pajusalu.

**Data curation:** Mihkel Pajusalu, Iaroslav Iakubivskyi, Olli Knuuttila.

**Formal analysis:** Gabriel Jörg Schwarzkopf.

**Funding acquisition:** Mihkel Pajusalu.

**Investigation:** Iaroslav Iakubivskyi, Gabriel Jörg Schwarzkopf, Olli Knuuttila, Maximilian Bührer, Mario F. Palos, Hans Teras, Andris Slavinskis.

**Methodology:** Mihkel Pajusalu, Iaroslav Iakubivskyi, Gabriel Jörg Schwarzkopf, Timo Väisänen, Maximilian Bührer, Hans Teras.

**Project administration:** Andris Slavinskis.

**Software:** Mihkel Pajusalu, Iaroslav Iakubivskyi, Gabriel Jörg Schwarzkopf, Olli Knuuttila, Timo Väisänen, Maximilian Bührer, Mario F. Palos, Hans Teras.

**Supervision:** Mihkel Pajusalu, Jaan Praks, Andris Slavinskis.

**Validation:** Iaroslav Iakubivskyi, Olli Knuuttila, Guillaume Le Bonhomme, Jaan Praks, Andris Slavinskis.

**Visualization:** Iaroslav Iakubivskyi, Olli Knuuttila, Timo Väisänen, Maximilian Bührer, Mario F. Palos, Guillaume Le Bonhomme.

**Writing – original draft:** Iaroslav Iakubivskyi.

**Writing – review & editing:** Mihkel Pajusalu, Iaroslav Iakubivskyi, Olli Knuuttila.

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
