## [Decision Letter · Decision Letter 0]

11 Aug 2021

PONE-D-21-15766

SISPO: Space Imaging Simulator for Proximity Operations

PLOS ONE

Dear Dr. Iakubivskyi,

Thank you for submitting your manuscript to PLOS ONE. After careful consideration, we feel that it has merit but does not fully meet PLOS ONE’s publication criteria as it currently stands. Therefore, we invite you to submit a revised version of the manuscript that addresses the points raised during the review process.

Please find below the reviews for your paper.

We look forward to receiving your revised manuscript.

Kind regards,

Antonio Agudo

Academic Editor

PLOS ONE

Journal Requirements:

The research for this article was partly supported by the University of Tartu ASTRA project 467

2014–2020.4.01.16-0029 KOMEET “Benefits for Estonian Society from Space Research and 468

Application”, financed by the EU European Regional Development Fund

1. Mihkel Pajusalu

2. PUTJD601 and MOBTP151

3. Estonian Research Council

4. https://www.etag.ee/en

5. he funders had no role in study design, data collection and analysis, decision to publish, or preparation of the manuscript.

3. Please ensure that you refer to Figure 1 in your text as, if accepted, production will need this reference to link the reader to the figure.

4. We note that Figures 1, 4, 5, 6, 8, 12, 13 and 14 in your submission contain copyrighted images. All PLOS content is published under the Creative Commons Attribution License (CC BY 4.0), which means that the manuscript, images, and Supporting Information files will be freely available online, and any third party is permitted to access, download, copy, distribute, and use these materials in any way, even commercially, with proper attribution. For more information, see our copyright guidelines: http://journals.plos.org/plosone/s/licenses-and-copyright.

a. You may seek permission from the original copyright holder of Figures 1, 4, 5, 6, 8, 12, 13 and 14 to publish the content specifically under the CC BY 4.0 license. 

Additional Editor Comments:

Dear authors,

The submission was reviewed by two experts in the field. Both of them pointed out some issues with the current submission. For instance, a lack of a properly experimental evaluation (R1) after comparing the generated and rendered images with respect to ground truth. This is really relevant to validate the work, especially considering the other reviewer (R2) considers the paper is just a description of capabilities of the software. In addition to that, the real contribution (R2) should be clarified, and add some missing details in the document. Considering that, the authors should carefully improve the document following point by point all the reviewers’ comments, especially the provided by R1 regarding the comparison and validation of the method. Unfortunately, I cannot consider the paper for publication without before including all the previous improvements.

Best

Reviewers' comments:

Reviewer's Responses to Questions

**Comments to the Author**

1. Is the manuscript technically sound, and do the data support the conclusions?

Reviewer #1: Yes

Reviewer #2: Yes

2. Has the statistical analysis been performed appropriately and rigorously? 

Reviewer #1: No

Reviewer #2: N/A

3. Have the authors made all data underlying the findings in their manuscript fully available?

Reviewer #1: Yes

Reviewer #2: Yes

4. Is the manuscript presented in an intelligible fashion and written in standard English?

Reviewer #1: Yes

Reviewer #2: Yes

5. Review Comments to the Author

Reviewer #1: My main comment is that when generating simulated images, it would be expected to perform image comparison with real images vs the simulations to present meaningful data on the accuracy of the simulations, hence why a revision is recommended. There are also some undeveloped features that appear to be "add ons" that detract from the interesting and more detailed work that has been presented, by this I mean partially developed concepts (not justified or evaluated), e.g. displacement craters, spacecraft images etc. Other comments include:

Fig 1 compares four image simulators. This is misleading because the SurRender and PANGU images (A and B) appear to be from an unenhanced shape model, while (D) clearly has additional features added. You need to state the resolution of the shape model used for each (for any reasonable comparison) and whether additional features have been added (both SurRender and PANGU have this capability). Also, you should include comparison to a real Hayabusa image to show how they compare and then image statistical evaluations. This would show the unrealistic, flat-bottomed (???) small craters on image (D) which do not exist on the small rubble-pile type asteroid Itokawa.

The additional of height-mapped displaced features needs some further qualification and comparison to real images showing these features, and a discussion to explain how and where they could be realistically added to a scene, giving examples of a real images showing comparable features. Otherwise, this is misleading as the paper does not explain in what scenarios these features can be reasonably added. For example, the craters are unrealistic in the context (flat bottomed craters are generally large (e.g. >10km on the Moon, but this varies with gravity and surface density), the overlapping is also unrealistic in the context presented.

For the camera model distortion, why are Earth images used? Have the properties of a real camera used in a space mission been modelled and simulated (e.g. the AMICA camera that took images of asteroid Itokawa, used in many of your examples).

You compare simulated to real images in Fig 10 which is helpful, could you also do further statistical image analysis to evaluate how similar the images are to the real images from a computer vision perspective, i.e. running simple histogram analysis and other standard image-processing comparison techniques?

The spacecraft rendering (Fig 13) is interesting but there are very little details give such as model material properties supported, comparison with real images etc. If this is to be included in this paper, then this section needs further details.

The reconstruction section shown in Fig 14- Fig 18 is much better as it gives some information on the accuracy of the technique, but this approach is lacking in some of the earlier sections.

Reviewer #2: SISPO: Space Imaging Simulator for Proximity Operations

Authors present a framework to perform several tasks related to astrophysics field, such as asteroid orbit simulation, simulation of dust of comets, render the result images and corrections on camera optical aberrations. For most of the tasks, the authors use third party software that is wrapped and prepared to be used in a more easy manner.

The article is just a description of the capabilities of the software. This is not a research paper.

The software is publicly available with limited documentation, which makes it not so straightforward to use, but still more easy than use the third party software separately.

In the current state:

1. If journal only accepts research papers: Reject

2. Otherwise: Minor revision

The document needs to be improved. It is very strange to find all figures at the end of the document, making the reader constantly jump up and down.

The contributions of the authors are not so clear as many of the capabilities come from a third party software, and these capabilities are not integrated into a single software but need to be run separately, but still can be useful software for the community.

Minor comments:

Table 1, mentioned in page 3, but appears in page 8. Table 1 page 3 should be different table, which I can't find.

line 244, not sure what authors mean by the "the like"

line 239 Cycle -> Cycles

6. PLOS authors have the option to publish the peer review history of their article (what does this mean?). If published, this will include your full peer review and any attached files.

Reviewer #1: No

Reviewer #2: No

---

## [Author Response · Author response to Decision Letter 0]

11 Oct 2021

Response to Reviewers PONE-D-21-15766

SISPO: Space Imaging Simulator for Proximity Operations

24.09.2021

Dear all,

Thank you for revising our manuscript and the excellent comments you have provided; this is precious information and helps us improve our paper. One general remark is that we developed this tool to simplify our work on the OPIC instrument development for the ESA Comet Interceptor mission (among other instruments we investigate). We found this tool very useful for the actual payload developed that will be launched with the Ariel telescope in 2029, and we thought to make this tool public for others to use with supporting paper describing it. According to the journal's statement: "PLOS ONE considers Research Article submissions that report new methods, software, databases and tools as the primary focus of the article". Please find responses to addressed issues below, as well as marked changes in the attached file.

Reviewer #1: My main comment is that when generating simulated images, it would be expected to perform image comparison with real images vs the simulations to present meaningful data on the accuracy of the simulations, hence why a revision is recommended. There are also some undeveloped features that appear to be "add ons" that detract from the interesting and more detailed work that has been presented, by this I mean partially developed concepts (not justified or evaluated), e.g. displacement craters, spacecraft images etc. Other comments include:

Thank you for the comment. Indeed it is valuable to include comparison for the produced images. We have included two new comparisons in (new) Figure 5 (OpenGL vs AMICA) and Figure 6 (Cycles plain vs AMICA). Cycles procedural texturing image is not reasonable to compare with the original image since it introduces new features and displaces the surfaces. It is not meant to reproduce the exact images that already exist but provides an opportunity to create a versatile set of new images using the same model: this is crucial for developing autonomous operations and training the algorithm to make the right decisions. 

Fig 1 compares four image simulators. This is misleading because the SurRender and PANGU images (A and B) appear to be from an unenhanced shape model, while (D) clearly has additional features added. You need to state the resolution of the shape model used for each (for any reasonable comparison) and whether additional features have been added (both SurRender and PANGU have this capability). Also, you should include comparison to a real Hayabusa image to show how they compare and then image statistical evaluations. This would show the unrealistic, flat-bottomed (???) small craters on image (D) which do not exist on the small rubble-pile type asteroid Itokawa.

Thank you for the comment. The same input model is used in all of the simulators. This table demonstrates the overview of the SISPO and does not intend to show the SISPO advantages over other simulators; it affirms the capabilities of the same model: each simulator is useful but uses different approaches, and we wanted to introduce them and provided relevant references for the reader. We have clarified new features generations in the text and table; hopefully, it is clearer now.

The additional of height-mapped displaced features needs some further qualification and comparison to real images showing these features, and a discussion to explain how and where they could be realistically added to a scene, giving examples of a real images showing comparable features. Otherwise, this is misleading as the paper does not explain in what scenarios these features can be reasonably added. For example, the craters are unrealistic in the context (flat bottomed craters are generally large (e.g. >10km on the Moon, but this varies with gravity and surface density), the overlapping is also unrealistic in the context presented.

Thank you for the comment. This is a user-defined decision and specific need. This tool has a particular set of capabilities that can be applied to different sizes of bodies, and we only show what is possible. While your comment about craters is valid based on the observed families of asteroids, the simulator user can adjust these sizes to their demands. For example, as you can see in Fig 8(B) these features are minimally used.

For the camera model distortion, why are Earth images used? Have the properties of a real camera used in a space mission been modelled and simulated (e.g. the AMICA camera that took images of asteroid Itokawa, used in many of your examples).

We have removed these images due to conflict with copyright as requested by the editor. The distortions of AMICA is so small that other errors become more significant in the analysis; therefore, we did not include them. The text provides a reference to the thesis that presents very detailed examples and capabilities of the SISPO's OASIS package in Bührer, 2020.

You compare simulated to real images in Fig 10 which is helpful, could you also do further statistical image analysis to evaluate how similar the images are to the real images from a computer vision perspective, i.e. running simple histogram analysis and other standard image-processing comparison techniques?

Thank you for the comment. We have included the comparison.

The spacecraft rendering (Fig 13) is interesting but there are very little details give such as model material properties supported, comparison with real images etc. If this is to be included in this paper, then this section needs further details.

Thank you for the comment. This article focuses mainly on small solar system bodies. As the use case, we have shown additional applications. A description of all underlying parameters and shaders would be misleading and occupy the big part of the paper with shifted focus. This example is listed because it can also be used for developing formation-flying algorithms and orbital proximity operations; we have added it to the text.

The reconstruction section shown in Fig 14- Fig 18 is much better as it gives some information on the accuracy of the technique, but this approach is lacking in some of the earlier sections.

Thank you, indeed this section is demanding numerical comparison.

Reviewer #2: SISPO: Space Imaging Simulator for Proximity Operations

Authors present a framework to perform several tasks related to astrophysics field, such as asteroid orbit simulation, simulation of dust of comets, render the result images and corrections on camera optical aberrations. For most of the tasks, the authors use third party software that is wrapped and prepared to be used in a more easy manner.

The article is just a description of the capabilities of the software. This is not a research paper.

The software is publicly available with limited documentation, which makes it not so straightforward to use, but still more easy than use the third party software separately.

Thank you for your comment. According to the PLOS ONE statement "PLOS ONE considers Research Article submissions which report new methods, software, databases and tools as the primary focus of the article." Source: https://journals.plos.org/plosone/s/what-we-publish

In the current state:

1. If journal only accepts research papers: Reject

2. Otherwise: Minor revision

The document needs to be improved. It is very strange to find all figures at the end of the document, making the reader constantly jump up and down.

Thank you for the comment but that is the requirement of this journal. The PLOS ONE requirements: the figures are not embedded in the latex file and must be uploaded separately, which end up at the end of the automatically generated file.

The contributions of the authors are not so clear as many of the capabilities come from a third party software, and these capabilities are not integrated into a single software but need to be run separately, but still can be useful software for the community.

Thank you for the comment. All the packages are included in a single pipeline, as visualised in Fig 2. The authors wrote all the scripts and performed simulations and analyses of the software. 

Minor comments:

Table 1, mentioned in page 3, but appears in page 8. Table 1 page 3 should be different table, which I can't find.

Thank you for pointing it out. There was a table in the beginning but later was represented as Figure 1. I have changed it in the text, it should be Fig 1.

line 244, not sure what authors mean by the "the like"

It was removed.

line 239 Cycle -> Cycles

It is corrected.

---

## [Decision Letter · Decision Letter 1]

31 Jan 2022

SISPO: Space Imaging Simulator for Proximity Operations

PONE-D-21-15766R1

Dear Dr. Iakubivskyi,

We’re pleased to inform you that your manuscript has been judged scientifically suitable for publication and will be formally accepted for publication once it meets all outstanding technical requirements.

Kind regards,

Antonio Agudo

Academic Editor

PLOS ONE

Additional Editor Comments (optional):

Thank you for providing a review point by point. The paper is ready for publication.

Reviewers' comments:

Reviewer's Responses to Questions

**Comments to the Author**

1. If the authors have adequately addressed your comments raised in a previous round of review and you feel that this manuscript is now acceptable for publication, you may indicate that here to bypass the “Comments to the Author” section, enter your conflict of interest statement in the “Confidential to Editor” section, and submit your "Accept" recommendation.

Reviewer #2: All comments have been addressed

Reviewer #3: All comments have been addressed

2. Is the manuscript technically sound, and do the data support the conclusions?

Reviewer #2: Yes

Reviewer #3: Yes

3. Has the statistical analysis been performed appropriately and rigorously? 

Reviewer #2: N/A

Reviewer #3: Yes

4. Have the authors made all data underlying the findings in their manuscript fully available?

Reviewer #2: Yes

Reviewer #3: Yes

5. Is the manuscript presented in an intelligible fashion and written in standard English?

Reviewer #2: Yes

Reviewer #3: Yes

6. Review Comments to the Author

Reviewer #2: Based on the comments of the authors regarding my main concerns about journal format and topic, I have no opposition to accept the paper.

Also, authors have included comparisons between real/simulated images which makes the document more interesting for a general readers.

Reviewer #3: Authors have addressed all the comments in the review. There are not more major concerns. Thank you very much.

7. PLOS authors have the option to publish the peer review history of their article (what does this mean?). If published, this will include your full peer review and any attached files.

Reviewer #2: No

Reviewer #3: No

---

## [Editor Report · Acceptance letter]

21 Feb 2022

PONE-D-21-15766R1 

SISPO: Space Imaging Simulator for Proximity Operations 

Dear Dr. Iakubivskyi:

I'm pleased to inform you that your manuscript has been deemed suitable for publication in PLOS ONE. Congratulations! Your manuscript is now with our production department. 

Kind regards, 

on behalf of

Dr. Antonio Agudo 

Academic Editor

PLOS ONE